# What is the impact of targeted therapies given within phase I trials on the cognitive function of patients with advanced cancer: a mixed-methods exploratory study conducted in an early clinical trials unit

Sarah Stapleton [1,2] Anne-Sophie Emma Darlington,[2] J S de Bono,[1,3] Theresa Wiseman[2,4]

¹Drug Development Unit, Royal Marsden Hospital Sutton, Sutton, UK
²Faculty of Health Sciences, University of Southampton, Southampton, UK
³Institute of Cancer Research Division of Cancer Therapeutics, London, UK
⁴The Royal Marsden NHS Foundation Trust, London, UK

**Correspondence to**
Sarah Stapleton;
Sarah.Stapleton@rmh.nhs.uk

## ABSTRACT

**Introduction** Novel therapies such as small protein molecule inhibitors and immunotherapies are tested in early phase trials before moving to later phase trials and ultimately standard practice. A key aim of these clinical trials is to define a toxicity profile, however, the emphasis is often on safety with measurements of organ toxicity. Other subjective side effects can be under-reported because they are not measured formally within the trial protocols. The concern from clinical practice is that cognitive toxicity is poorly studied and may be under-reported in this context. This could lead to toxicity profiles of new treatments not being fully described and patients with unmet need in terms of acknowledgement and support of symptoms. This protocol outlines a framework of an exploratory study with feasibility aspects to investigate the impact and experience of cognitive changes for patients on phase I trials.

**Methods and analysis** This is a mixed-methods study, combining quantitative and qualitative approaches. The sample is 30 patients with advanced cancer who are participating in phase I trials of novel therapies in the early clinical trials unit of a specialist cancer centre. A test battery of validated cognitive assessments will be taken alongside patient reported outcome measures at three time points from baseline, day eight and day 28 post start of treatment. At day 28, a semi-structured interview will be conducted and the narrative thematically analysed. Results will be integrated to offer a comprehensive description of cognitive function in this patient group.

**Ethics and dissemination** The study has received full HRA and ethical approval. It is the first study to introduce formal cognitive assessments in a cancer phase I trial context. The study has the potential to highlight previously unreported side effects and more importantly unmet need in terms of care for patients who are participating in the trials.

## INTRODUCTION

In cancer drug development advances in the understanding of genetics and the

---

### STRENGTHS AND LIMITATIONS OF THIS STUDY

⇒ This is the first study to investigate cognitive function longitudinally during the course of phase I trials.
⇒ The findings of the trial will establish feasibility of conducting formal and consistent cognitive testing in early clinical drug trial protocols.
⇒ Describing the symptoms of patients in this group will help to identify unmet need in terms of support and management.
⇒ Recommendations of the International Cancer and Cognition Task Force to harmonise research in this field were adopted in the trial design.
⇒ Due to the exploratory nature and feasibility aspects of the study, the sample size is not statistically powered. A future study would benefit from a powered calculated sample size to demonstrate any treatment related change with more certainty.

---

biochemical pathways involved in cancer have helped researchers to rationally design 'molecularly targeted drugs'. These novel targeted therapies and immunotherapies are first tested clinically in patients within phase I trials in cancer centres. Due to the experimental nature of phase I trials and the fact that little is known about the toxicities, patients need to have exhausted all standard treatment options to participate. To contextualise, these patients will generally have advanced cancer and pretreated disease. It is a heterogeneous patient population in terms of tumour types. However, what the patients do have in common is a shared experience of the continuous cycle of treatment and disease progression to the point where there is no further treatment available. Patients on phase I trials often have significant tumour burden with metastatic disease and a life

**BMJ**

limiting prognosis. These patients remain motivated to participate in phase I trials of experimental therapies and despite having advanced progressive disease the criteria for participating in phase I means the patients do have to have a certain level of fitness to meet eligibility requirements of specific protocols.

At this stage of drug development, the main aims are to find the optimal dose, investigate the pharmacology and to define the toxicity profiles.[1] When considering toxicity profiles of these potential new treatments the emphasis is on safety and this is reflected in many of the standard assessments in the protocols such as cardiology monitoring,renal and liver function tests.[2] Clinicians and nurses involved in the conduct of phase I trials do perform medical history taking and nursing assessments to gain a full perspective of other side effects reported but they are interpreted and recorded in a subjective manner usually using the Common Terminology Criteria for Adverse Events (CTCAE) which is a National Institute of Health-National Cancer Institute, Division of Cancer Treatment and Diagnosis, Cancer Therapy Evaluation Programme initiative. It is a multidimensional tool designed specifically for use in clinical trials. The tool enables the clinician to categorise symptoms that are graded 0–5, translating respectively to none, mild, moderate, severe, life-threatening and death. The use of the CTCAE is commonplace in all phase I trials.[3]

This exploratory study investigates the impact of targeted therapies within the context of an experimental trial on the cognitive function of patients with advanced cancer. It is suspected that some of the targeted therapies tested within phase I trials may have a negative impact on the patient's cognitive function. Evidence from clinical practice suggests that patients are experiencing symptoms relating to cognitive function, however this does not seem to translate to reports of cognitive toxicity within clinical trial data. As cognitive function is not formally assessed the extent, nature or domain of cognitive impairment (CI) is not well described. There is also suggestion in literature that the difficulty in ascertaining the relationship between CI and targeted therapies may be due to the under-reporting of cognitive symptoms in clinical trials.[4 5]

## Rationale

Impairment of cognitive function in the field of traditional cytotoxic chemotherapy has been investigated and it is known that such a symptom impairs quality of life.[6] Therefore, if cognitive function was investigated and described in the context of patients on phase I trials, it could provide a basis of knowledge on which to develop interventions and coping strategies with the aim of preventing or ameliorating the effects of this possible decline in cognitive function.

Cognitive function is described as the internal processes involved in making sense of the world and the capacity to make decisions.[7] Such processes include attention, perception, learning, memory, language, problem solving, reasoning and thinking.[8] These functions are defined

in terms of domains that can broadly be divided into sensory and motor skills, memory, attention and concentration, executive function (which includes reasoning and problem solving) and language and verbal skills.[9] While the domains overlap in terms of function each can be assessed using validated cognitive assessment tools. Impaired cognitive function in patients with cancer can present as loss of memory, difficulty with word finding, loss of concentration and difficulty in task switching.[10] Even though cognitive function is not assessed as part of the phase I trials, anecdotal reports from some patients suggest that they experience some deterioration while on the trial, which is often temporal to dosing of drug.

For patients this has implications in that they may have a symptom they feel unable to discuss. This in turn can mean that they do not receive any help or support regarding the problem which can be isolating and frightening for the patient.[11] Not having the symptom acknowledged or having no information to warn about the possibility of it happening has been reported as being distressing and difficult for patients.[11 12]

Implications on a broader scale will mean that better defined toxicity profiles will improve understanding of consequences of treatments. This will assist in developing interventions that help patients make informed choices about having certain treatments and be better equipped to cope with the consequences of them. This may be as simple as recognition of the potential problem with information and preparation for the patient or the implementation of strategies to help with symptoms such as compensatory memory techniques, rehabilitation and relaxation.[13 14]

In terms of relevance the numbers of patients initially receiving such treatments in phase I trials may be relatively small, usually tens to hundreds. Nevertheless, for these patients with a limited prognosis issues regarding side effects are important. Once a drug has gone through the phases of testing and is licensed for practice the numbers of patients receiving these treatments could then become thousands.[15] The number of patients with cancer participating in clinical trials is increasing.[16] Therefore, it is important to find out accurately what is happening to patients on clinical trials so they can be supported with the management of side effects such as diminished cognitive function. In order to begin to address this and improve the care and understanding of patients on phase I trials this proposed study will be the first exploration using formal assessment of cognitive function in this context.

## Evidence gap

Early work in chemotherapy related CI highlighted a disparity in trial designs and some limitations in research in this field, such as the use of cross-sectional trial design offering no baseline from which to measure change. Lack of consistency in the type of cognitive assessment methods used in the trials also lead to broad differences in reporting of the prevalence of CI. This led to a team of expert cognitive researchers founding a multidisciplinary

group known as the International Cancer and Cognition Task Force (ICCTF). The ICCTF was founded in 2004 and one of the aims was to produce guidelines to harmonise study design. The recommendations were to include pretreatment assessment, longitudinal follow-up, the use of a control group and a core set of cognitive assessment tests when investigating cognitive function in patients with cancer.[17] All recommendations will be adopted for this protocol except the use of a control group. It would be difficult to match a control for this patient group as the main reason that patients do not enter a phase I trial is due to a deterioration in clinical condition. Therefore, patients in a comparable situation would likely be receiving palliative care only and it may not be appropriate to ask them to participate as a control.

The subsequent studies that followed these guidelines continued to show associations between systemic chemotherapy treatment and CI as well as discovering that patients with cancer often have a degree of CI at baseline when compared with matched healthy controls.[18–20] The studies also describe the manifestation of CI as a decline in concentration, memory, verbal fluency and executive functioning, which includes higher order cognitive abilities of planning and abstract thinking. Due to the longitudinal design some have been able to demonstrate compelling evidence of dose dependent relationships between treatment and severity of diminished cognitive function.[21–25]

Some of the studies also highlighted the value of collecting the patient's own perception of cognitive function as they showed that this can vary from the validated cognitive assessment measures,.[26 27] What is not addressed in many of these studies is the meaning to the patient in terms of how it may affect their lives. Just a small number of qualitative studies have been conducted exploring the impact of cognitive function on the quality of life of patients. They show that there is a negative effect on work, being unable to return to previous occupations and detrimental effects on emotional, social and psychological aspects of daily life.[28]

A scoping literature review was conducted with the aim of understanding the status of assessment and reports of cognitive function within phase I trials. The scoping review used the Preferred Reporting Items for Systematic Reviews and Meta-Analyses extension for Scoping Reviews checklist as a framework.[29] Papers were included if they reported cognitive symptoms or symptoms categorised as cognitive within the toxicity reports of the trial. The purpose of this was to identify any issues relating to cognitive function, including how it was assessed and reported. Also included were any studies investigating cognitive function in a phase I population. Other observational studies that explored symptom presentation and reporting in phase I trials were included. The purpose of this was to understand any factors that are specific to this patient population and context that may influence identification of cognitive symptoms. Excluded were studies that did not include cognitive issues in the reports. Studies of patients with central nervous system tumours and paediatric patients were excluded as due to the nature of the disease and treatment there are cognitive issues specific to each of these patient's groups. Search limits included papers in English and dates of publication 2010–2021.

Just one study in the review specifically investigated cognitive function in a population of phase I patients. The aim was to identify the relationship between cognitive function and intact decisional capacity in relation to informed consent. The cross-sectional design assessed the cognitive function of patients at baseline prior to starting the trial so there was no longitudinal assessment or feasibility assessment throughout the course of the clinical trial. The sample was representative of a phase I population, outcome measures were valid, and associations of confounding variable were adjusted for in the analyses. The research team found that a proportion of patients had CI at baseline and those with worse cognitive function were less likely to understand and remember some of the key principles of the trial. The study was limited due to the cross-sectional design and the researchers do propose that further longitudinal work assessing cognitive function in patients on phase I trials would be of value.[30]

Evidence of cognitive toxicity reports in the phase I trials was limited. A small number did report cognitive toxicity. The symptoms were not well characterised and sometimes aggregated or reported as a cognitive category rather than the symptom, making it difficult to understand exactly what the adverse events were.

A phase 1 trial investigating safety and efficacy of BLU-285, a highly selective targeting inhibitor of mutant KIT and PDGFR in a population of 116 patients with Gastro-Intestinal Stromal Tumour reported CI as a toxicity. The symptoms are aggregated in the report as cognitive effects. The paper reports that 46% (n=115) have all grade cognitive effects and group together memory impairment, confusional state, cognitive disorder and encephalopathy within this category.[31] This perhaps represents a lack of clarity in working definitions of CIs.

Other reports of cognitive toxicity within phase I include a combination study investigating the maximum tolerated dose and toxicity profile of a Vascular Endothelial Growth Factor inhibitor and Pi3K inhibitor, for patients did not include cognitive assessments but did include specific mental health diagnostic tools in the trial design. Despite the lack of cognitive assessments 10% (n=30) of patients on the trial reported cognitive disturbance. This is another example of the symptom being reported as a category rather than the actual symptom, hence the presentation of the cognitive disturbance is impossible to elucidate. This vague description leads to a poor understanding of the symptom which is attributed as a severe side effect of treatments. Indeed the cognitive disturbance was graded severe enough to be classified as a dose limiting toxicity. A dose limiting toxicity is a drug related adverse event predefined in the protocol and is significant as it may prevent further dose escalation. This means that the recommended doses could be affected.[32]

Despite not including any cognitive assessments within the trial protocol, a phase I trial of a Pi3K and mTor inhibitor in combination also reported 'cognitive disorders' in some patients on the trial this is further categorised as memory loss, confusional state, attention deficit and agitation. Causality was attributed to the investigational medicinal product.[33] Because no formal cognitive assessments were incorporated into the study design, the severity and duration of this is unclear.

None of the papers filtered in the review used formal cognitive assessments within the trial protocol, so the review confirms that there are no formal or consistent cognitive assessments as part of current standard practice within early clinical trials.

More recently studies are emerging that look at patient-reported outcome measures (PROMs) in phase I trials, in particular the PROMs version of the CTCAE (PRO-CTCAE) which is a multidimensional PROMs specifically designed by the NCI to capture symptomatic adverse events in clinical trials.[34] While its use is commonplace in phase 2 and 3 trials it's used less in early phase trials where definition of toxicity is a key endpoint.[35] One study recruited 265 patients on phase I trials and administered the PROMs at three time points. In the case of cognitive events there are two items on the scale of memory impairment and impaired concentration. The study team describe a difference between PROMs and clinician reports as 25.9% of patient reports of memory as compared with 0.5% of clinicians and in the case of impaired concentration 0.7%–28.1%, respectively. A similar study using the MD Anderson symptom inventory which is also a patient reported questionnaire focusing on 13 core common cancer symptoms.[36] It has one item relating to cognitive function labelled as 'difficulty remembering'. This study showed that up to 41% of patients identified as having difficulty remembering.

The number of experiences of CI in the PROMs studies are incongruous with the many papers filtered in this review where there are simply no or very limited reports of CI. This highlights potential unmet need in this patient group.

In summary, there is a lack of evidence to show the effects on cognitive function for patients on phase I clinical trials. This has a twofold consequence. First, that the side effects of the drugs may not be well characterised and only become apparent in latter stages of drug development or even when a drug becomes standard practice. Second patients who are participating in phase I trials may have unmet need in terms of identification and support of symptoms of CI. This research protocol outlines an exploratory framework for a study using mixed methods to both identify the existence of cognitive decline in these patients but also describe the impact this has on their everyday lives.

## Research question and aims

The study is not hypothesis testing and is framed as an exploratory study. The overarching aim is to identify, characterise and gain an understanding of the effects on the cognitive function of patients on novel targeted therapies given within the context of phase I trial. The study has an added feasibility aspect to gain an understanding of how acceptable and appropriate nurse led cognitive assessments are in this patient group bearing in mind the existing potential burden of the phase I trial assessments. The overarching research question for the study is:

'What is the impact of the targeted therapies given within the context of a phase I trial on the cognitive function of patients with advanced cancer?'

The research question can be broken down into six sub questions. The research sub questions are presented below aligned with the objectives, methods and analysis (table 1).

## METHODS AND ANALYSIS

This is a mixed-methods longitudinal study design using both qualitative and quantitative approaches with equal weighting.

## Sample

A non-probability sampling strategy technique of convenience sampling will be used. Most of the patients entering a phase I trial will be eligible for this study therefore this sampling technique will allow for the quickest access to patient recruitment. The patients will be recruited from the clinical trials unit within a specialist cancer hospital, where they attend on a weekly basis. The patient group is heterogeneous in terms of tumour type, age and gender. However, the similarities within the sample population will be the stage of disease, the types of previous treatment and context. Although a convenience sample will be used there will be some stratification in terms of gender. Due to sample size any stratification of other possible confounding factors such as age, educational status will not be possible. Age and educational status (as a key component of cognitive reserve) have been shown to have an influence of altered cognitive function. The sample split between male and female will aim to be equal, as it is known that there are cognitive differences relating to gender.[37] As this is an exploratory study with feasibility aspects, the sample size will be 30 patients.

## Inclusion criteria

- ► Patients with a confirmed diagnosis of cancer.
- ► Patients who are eligible for the trial and meet the trial specific eligibility criteria.
- ► Patients who are allocated to and commence a phase I trial (either IV or oral).
- ► Patients who are over eighteen.

## Exclusion criteria

- ► Patients with a primary central nervous system tumour.
- ► Patients who have received whole brain radiotherapy.
- ► Patients who are non-English speaking.
- ► Patients with uncontrolled psychiatric disorders.

**Table 1** Research questions aligned with objectives, methods and analyses

| Research question (RQ)) | Objectives | Methods of data collection | Analysis |
|---|---|---|---|
| RQ1: What is the cognitive function of the patients on Phase I trials of novel therapies | To describe the cognitive function over time of this patient group | Cognitive assessments of memory (Hopkins Verbal Learning Test-r) executive function (Comprehensive Trail Making test A and B) and verbal fluency (Controlled Oral Word Association Test) | Age adjusted T scores will be presented as means and SD with confidence intervals for each time point. Proportions of patients with CI will be presented at each time point. ANOVA repeated measures will be conducted to describe the change over the three time points. |
| RQ2: What is the experience of cognitive function and the impact of any deterioration on the lives of these people on Phase I trials? | To explore and understand the patient narrative surrounding cognitive function in this context. | Individual in-depth interviews PROMs of perceived CI Fact-cog | Thematic analysis (Braun and Clark) Decline at each time point will be present as predefined Minimum Clinically Important Differences (Cut-off points are available for perceived impairment subscale that can be used to present proportions of patients with impairment at each time point). |
| RQ3: Do patient's perspectives mirror cognitive outcome measures overtime? | To compare, contrast and look for confirmation, divergence and an in-depth characterisation of cognitive function in this patient group | Combined qual and quant | Integration at analysis stage using mixed-methods matrix to interpret findings |
| RQ4: What influence do potential confounding factors of depression, anxiety and fatigue have on the cognitive scores in this patient group? | To assess the impact of potential confounding variables on cognitive decline in this patient group. | PROMs of fatigue, anxiety and depression | ANOVA will be conducted to examine the influence of the confounding variables |
| RQ5: How acceptable are the cognitive assessments alongside the phase I clinical trial? | To measure acceptability of extra cognitive assessments alongside Phase I trials. | Recruitment, refusal and attrition rates recorded Qualitative enquiry as part of the interview to ask how acceptable the cognitive assessment battery has been | Measurements of uptake of trial, refusal to enter study (with reasons if possible) attrition rates (presented as numbers and percentages) Frequency of positive responses in qualitative questions Coding and thematic analysis of narrative response to enquiry about acceptability of questions. |
| RQ6: How accurate are nurse led cognitive assessments (the nurse has accredited training in Psychometric testing and is being supervised by a clinical psychologist). | | Measurement of no of approved accurate cognitive assessments undertaken as validated by psychologist. | Percentages of positive validation by supervising psychologist. |

ANOVA, analysis of variance ; PROMs, patient-reported outcome measures.

## Patient pathway

The following figure 1 describes the patient pathway. This is their schedule of assessments aligned with the phase I clinical trial pathway and schedule of assessments.

## Quantitative methods

This study follows the ICCTF recommendations of a core battery of validated cognitive assessments of memory, verbal fluency and executive functioning at baseline, day eight and day 28 poststart of trial treatment. As well as the cognitive assessments that will be conducted by the primary researcher, there will be PROMs of fatigue, depression and anxiety and self-perception of cognitive function. The methods of data collection and administration are summarised below (table 2).

## Qualitative methods

The qualitative method of data collection will be a semistructured interview which will take place on day 28. The interview will be digitally audio recorded and transcribed verbatim. The purpose of the interview is to drill down into issues identified by the quantitative assessments, but also to further elicit information regarding possible impact on the patient's life. The interviews are semistructured but can be flexible so that they can be guided by any issues highlighted by the quantitative data

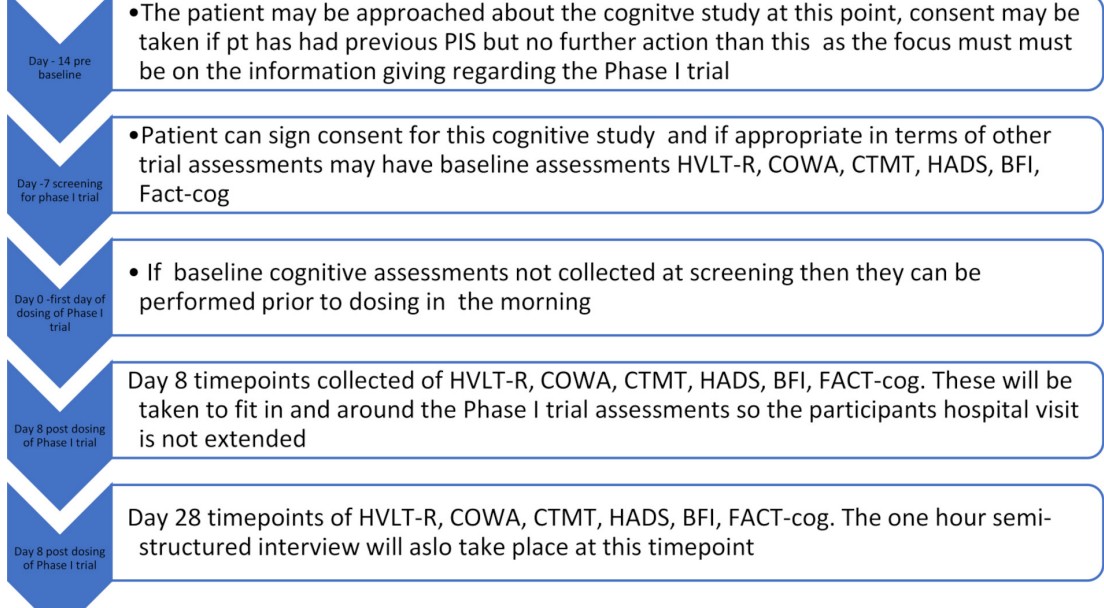

**Figure 1** Patient pathway aligned with phase I trial. BFI, Brief Fatigue Inventory; COWA, Controlled Oral Word Association; CTMT, Comprehensive Trail Making Test; HADS, Hospital Anxiety and Depression Scale; HVLT-R, Hopkins Verbal Learning Test-Revised;

collection. The interview schedule is in online supplemental appendix A.

The interview will be conducted by the primary researcher, in a private room and last for up to 1 hour. The participant will be told that they can have a break or stop the interview at any time. Interviewing patients on potentially sensitive issues can lead to an emotional response, either cathartic or distressing.[38] At the end of the interview there will be two questions regarding the experience of the cognitive assessments, this will provide information regarding the tolerability of performing such structured assessments in this patient group.

### Analysis plan

#### Research question one: are targeted therapies given within the context of phase I trials associated with deterioration in cognitive function

The aim of these analyses is to describe patterns of decline and explore the use of methods in this patient group. To answer the first research question and provide a description of the function of the three cognitive domains from baseline across the time points. The following analysis plan will be conducted:

The Hopkins Verbal Learning Test-Revised (HVLT-R), Comprehensive Trail Making Test (CTMT) A and B will be converted to age adjusted T scores using the professional manuals.[39–42] The Controlled Oral Word Association (COWA) will be converted to Z scores.

Sample means of the scores at each time points will be presented with SD and CIs. Box plots will be produced to view the data. The proportions of patients showing CI as described here will be reported as percentages. For the HVLT-R memory test CI will be defined as a T score of

less than 42.[43] For the CTMT A and B CI is defined as a T score less than 40 and for the COWA CI is defined as a Z score less than −1.[44]

To describe the change over time and comparison of means an analysis of variance (ANOVA) repeated measures test will be conducted. While caution should be taken to interpret these results due to sample size the ANOVA may help to explore possible trends in cognitive function and provide effect sizes for the changes in the dependant variables (cognitive scores). The results from the ANOVA analyses will also be used to answer research question 3, more detail will be given about the independent variables to be included in the model.

#### Research question 2: What is the experience of cognitive function and the impact of any deterioration on the lives of these people on phase I trials?

To answer the quantitative aspect of this question with the objective of assessing self-reported cognitive function the following analysis will be conducted. At each time point, the global score will be calculated according to the standard scoring protocol for FACT-cog.[45] The proportions of patients reporting perceived CI (PCI) on the PCI Subscale will be calculated at each time point using the cut-off point of>60.[46] Changes over time of PCI will be evaluated using the established MCID of 9 points difference from baseline.[47]

#### Research Question 3: What influence do potential confounding factors of depression, anxiety and fatigue have on the cognitive scores in this patient group?

To assess the association of potential confounding variables on cognitive decline in this patient group

**Table 2** Methods of data collection and administration

| Validated cognitive assessment tests | Process | Time taken to complete test |
|---|---|---|
| Controlled Oral Word Association[35] | This is an oral fluency test. The assessment is made by asking the participant to make verbal associations to letters of the alphabet by saying as many words as possible one can think beginning with a given letter. A sheet is available to record verbatim response. Total number of acceptable responses provides the raw score. | 5–10 min |
| Hopkins Verbal Learning Test-Revised[36] | This is a word list learning and memory test. Each form of the test consists of 12 nouns with four items taken from each semantic category. The word list is read to the patient who then must recall as many words as possible. The task is repeated two or three times. A 25 min interval is given after which a delayed recall test is performed. The tests are then scored in terms of total recall, delayed recall and retention %. | 40 min (including 25 min for delayed recall test) |
| Comprehensive Trail Making Test[37] | This test measures executive function, concentration and task switching. The test consists of a standardised set of visual search and sequencing tests. The raw score for each trail test is the number of seconds required for the participant to complete; time is added per error as the examiner corrects them. | 5–12 min |
| Patient-reported outcomes | Process | Time taken to complete test |
| Fact-Cog[38] | This is a 35-item questionnaire measuring the patient's own perception of cognitive function changes. It is based on a 5-point rating scale 'in the last 7 days time frame'. | 15 min |
| Hospital Anxiety and Depression Scale (HADS)[39] | The HADS is a 14-item scale for patients who may have other medical problems. It is essentially a screening tool for anxiety and depression. Seven items cover anxiety, and seven items cover depression. Each item has score of 0–3 making a potential score between 0 and 21 for each problem. | 15 min |
| Brief Fatigue Inventory[40] | The brief fatigue inventory is a six-item assessment tool measuring the patients' levels of fatigue. It records changes over short periods with a focus on fatigue in the previous 24 hours. | 5 min |
| Qualitative data collection | Process | Time taken to complete interview |
| Semi-structured interviews | The patient participates in a 1-hour semistructured interview. This takes place on the same day as the final objective measures. The questions explore the experience of any perceived cognitive issues as well as coping strategies and any other factors that may influence this experience. Within this interview there is a short section asking questions regarding acceptability of the objective measures as part of the feasibility aspect of this study. | 1 hour |

identified variables will be included in the ANOVA repeated measures. It is known there are some relationships between anxiety and depression, sex and fatigue with CI. It is also suggested in the literature that researchers investigating cognitive function should plan to account for these possible relationships.[48] Prior to inclusion in the ANOVA correlations between the dependant variables and independent variables will be run to ensure that the model will be data driven rather than theory driven.

Bivariate correlation between baseline DV and the IV will be conducted using the Pearson R standardised correlation. This will establish the variables required to be included in the ANOVA.

### Calculating effect size

Effect size analysis using Cohen's d will be conducted on all statistically significant changes from baseline to help understand the practical significance of the results. A small to moderate effect size is considered meaningful.

### Research questions 3 and 4

Proportions of recruitment, refusal and attrition rates of the study will be presented as percentages. A 70% accrual rate is selected as a level to demonstrate acceptability to patients.

Proportions of correctly completely and interpreted nurse administered cognitive assessments will be recorded as agreed by clinical psychologist reported as

overall number and percentages. It is expected that 90% agreement on cognitive scores to demonstrate accuracy.

## Qualitative data analysis

The digital audio recording will be transcribed verbatim. The data will be organised using the NVivo software package. The premise of thematic analysis is the examination and recognition of patterns within the data that are ultimately categorised into meaningful themes.[49] Thematic analysis can use different approaches and be applied to varying types of data, it may be theory driven or content driven. In this instance, it will be the content in the data that will drive the analysis.[50]

## Integration of data

At the stage of interpretation and reporting a mixed-methods matrix will be used as a joint display of findings. This is a suggested practical way of viewing and analysing the different types of data together and offers a tool to guide the narrative during reporting.[51 52] The comprehensive results and analysis will be held in a separate database.

## Patient and public involvement

The protocol has had patient and public involvement in the design and development stage. The protocol framework was presented and discussed at a Royal Marsden Hospital Patient and Carer Research Panel, PPI input led to decisions regarding timings and numbers of assessments. A patient representative was consulted and helped to develop the questions for the semistructured interview.

## Ethics and dissemination

The study has obtained Research Ethics Committee approval in the UK reference number: 18/LO.0084 and full Health Research Authority approval IRAS project number: 235668. The study is currently open to recruitment after a long pause in recruitment due to the COVID-19 pandemic.

**Acknowledgements** The author would like to acknowledge the Royal Marsden Patient and Carer Research Review Panel for their contribution to the protocol development.

**Contributors** SS: primary researcher, research design concept, protocol development and writing. A-SED: PhD primary supervisor, protocol development. JSdB: clinical oversight of the study as head of unit, review of protocol development. TW: principal investigator, PhD supervisor, protocol development.

**Funding** The primary researcher's post is funded by Cancer Research UK.

**Disclaimer** The views expressed in this article are those of the author(s) and not necessarily those of the NHS, the NIHR or the Department of Health.

**Competing interests** JSdB has served on advisory boards and received fees from many companies including Astra Zeneca, Astellas, Bayer, Bioxcel Therapeutics, Boehringer Ingelheim, Cellcentric, Daiichi, Eisai, Genentech/Roche, Genmab, GSK, Janssen, Merck Serono, Merck Sharp & Dohme, Menarini/Silicon Biosystems, Orion, Pfizer, Qiagen, Sanofi Aventis, Sierra Oncology, Taiho, Vertex Pharmaceuticals. He is an employee of The ICR, which have received funding or other support for his research work from AZ, Astellas, Bayer, Cellcentric, Daiichi, Genentech, Genmab, GSK, Janssen, Merck Serono, MSD, Menarini/Silicon Biosystems, Orion, Sanofi Aventis, Sierra Oncology, Taiho, Pfizer, Vertex, and which has a commercial interest in abiraterone, PARP inhibition in DNA repair defective cancers and PI3K/

AKT pathway inhibitors (no personal income). He was named as an inventor, with no financial interest, for patent 8 822 438. JSdB is a National Institute for Health Research (NIHR) Senior Investigator.

**Patient and public involvement** Patients and/or the public were involved in the design, or conduct, or reporting, or dissemination plans of this research. Refer to the Methods section for further details.

**Patient consent for publication** Not applicable.

**Provenance and peer review** Not commissioned; externally peer reviewed.

**ORCID iD**
Sarah Stapleton http://orcid.org/0000-0002-1579-9447

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
