## [Reviewer comments · BMJ Open]

ARTICLE DETAILS

TITLE (PROVISIONAL)	What is the impact of targeted therapies given within Phase I trials on the cognitive function of patients with advanced cancer. A mixed methods exploratory study conducted in an early clinical trials unit.
AUTHORS	Stapleton, Sarah; Darlington, Anne-Sophie; de Bono, J. S.; Wiseman, Theresa

VERSION 1 – REVIEW

REVIEWER	Horowitz, Todd National Institutes of Health, National Cancer Institute
REVIEW RETURNED	14-Jun-2021

GENERAL COMMENTS	This protocol describes a study aimed at measuring and characterizing the effects of novel targeted therapies on the cognitive function of cancer patients enrolled in a Phase I trial. The study will combine quantitative (e.g., neuropsychological tests, PRO questionnaires) and qualitative (structured interview) data to come to a complete picture of the cognitive challenges facing these patients. I think the study is timely and will fill an important gap in our evidence about cognitive function in cancer patients. The study lacks a control group and will use a convenience sample; both of these decisions are justified in the context of the Phase I trial. However, I think the selection of primary and secondary endpoints does require additional explanation. Please explain why the HVLt was chosen as the primary endpoint, over the other objective neuropsychological tests. Also, why is change from baseline to day 8 a secondary and not primary endpoint? Table 1: SQ1 question is unfinished Random odd capitalization choices throughout the manuscript, e.g. on p. 5 “In Cancer drug...”, table 1: “...in Psychometric testing...” p. 11, Qualitative methods: I hope that by “illicit information” you mean “elicit information” p. 12 “The primary endpoint will be analysed in the first instance using a paired t-test if the data are normally distributed, if not, a non-parametric method such as the Wilcoxon matched pairs signed rank test will be used.” Consider using the Bayesian t-test, which allows you to compare evidence for and against the null hypothesis.
--

REVIEWER	DeAngelis, Carlo
-----------------	------------------

	Sunnybrook Health Sciences Centre, Pharmacy
REVIEW RETURNED	18-Aug-2021

GENERAL COMMENTS	I found your research idea of interest and believe that there is a need to more effectively monitor patients enrolled in clinical trials for side effects, not just for cognitive dysfunction but for other side effects as well. There needs to be a more proactive rather than the current reactive approach which relies on the patient to report side effects sometimes weeks after receiving therapy. Overall the mixed methods approach is appropriate. I do have some concerns listed below. It may be that you have already thought of the issues I outline but have not incorporated the detail in the manuscript. However, I can only comment on what I have available to me to read. I structured my comments in the order in which they arose during the reading of the manuscript. These include minor edits and more substantive questions/concerns. Page 4 - Lines 34/35 – end of sentence – I believe “for” should be deleted, for the sentence to read properly. Page 4 - Lines 37/38 – beginning of the sentence – I believe “is” should be deleted, for the sentence to read properly Page 4 - Line 59/60 – mid-sentence – “form” should be “from” Page 5 – Line 15 – mid-sentence – “to” should be “do” Page 5 – Lines 38-41 – Sentence re: CTCAE – the Common Terminology Criteria for Adverse Events (CTCAE) is a National Institutes of Health – National Cancer Institute, Division of Cancer Treatment and Diagnosis, Cancer Therapy Evaluation Program initiative not an EORTC initiative. Please correct the sentence with respect to the source of the CTCAE. Also use the correct designation: Common Terminology Criteria for Adverse Events (CTCAE). Finally re: citation #3 – The link does not work. Rather than direct the reader to a PDF version of the NCI-CTCAE suggest using the main page link: https://ctep.cancer.gov/protocoldevelopment/electronic_applications/ctc.htm Page 5 – Lines 47-49 – Sentence – “This exploratory study investigates the impact on the cognitive function of these patients with advanced cancer, who are participating in Phase I trials.” Grammatically is missing the impact of “what”. Suggest: “This exploratory study investigates the impact of novel targeted therapies on the cognitive function of patients with advanced cancer, who are participating in Phase I trials.” Page 7 – Lines 47-54 – This is a reasonable justification as to why a group of control patients will not be recruited into the study. I would also suggest that given that this is an exploratory study, not having a control group is also consistent with the study’s objectives. The ICCTF also suggests that a control group is not always feasible and that the control group will depend on the research question under study. Page 8 – Line 5 – re: citing of statement made in the sentence – “19,20,21,22,23,24,25,26” - Multiple citations that are consecutively numbered can be cited as a range i.e. 19-26 Page 8 Lines 32-37 – re: Definition of Dose Limiting Toxicity – Please revise the sentence to include that the occurrence of a DLT prevents further dose escalation of the study drug. See NIH-NCI definition at: https://www.cancer.gov/publications/dictionaries/cancer-terms/def/dose-limiting Page 8 – Lines 37-57 - I could not verify the information cited (references
--

30-33).

- Reference # 30 (Bendell JC, et al.) – BKM120 was given as a single agent not in combination with an mTOR inhibitor as suggested in the manuscript text. In addition in the Bendell paper, I could not find any mention in the text of “memory loss, confusion, or attention deficit and agitation”. They reported mood effects and anxiety. Terms included in mood effects, included “mood altered, anxiety, depression, emotional disorder, crying episodes, hallucinations and irritability”. The paper reports all Grade “Mood altered” as 20%, (Table 2). Anxiety was reported separately (17% Table 2). In Table 2 the study population size was 35, not 28 as suggested in this manuscript. I did not have access to the supplemental material for the Bendell paper where there may have been corroborating information supporting your statement re” the Bendell paper. I am also concerned with the use of “Mood Effects” being conflated as cognitive dysfunction. This is not consistent with your definition/description of cognitive dysfunction on Page 6 of the manuscript (mid-page).

- Reference #31 – (Fong PC, et al.) – I could not find any mention of mood effects or any other CNS effects in the toxicity table (Table 3). In text they describe two cases of DLT due to altered mood and fatigue (1 patient) and another patient with extreme somnolence. Here again I am concerned about conflating the mentioned side effects with cognitive dysfunction.

- Reference # 32 – (McKay RR, et al.) In this paper they appear to make a distinction between “cognitive disturbance” and “mood effects”. They used CTCAE v 4.0 to define this endpoint. There was one patient who had a DLT related to cognitive dysfunction. This paper is your best example of what you are claiming, i.e. cognitive dysfunction is not systematically assessed and because assessment relies on patient reporting the adverse effect, it may be missed. I also note that in this paper baparrisib (= BKM120 from citation # 30) is used in combination. Did you mean to refer to this paper earlier in the paragraph when citing reference #30 as a study of BKM120 given in combination?

- Reference #33 – (Heinrich MC, et al. Abstract) – There is no mention of mood effects or cognitive dysfunction.

Overall I found this paragraph to not accurately reflect the information in the publications cited. I do not disagree with your assertion that cognitive dysfunction is not commonly reported in Phase I trial. I do however, have some concern that the publications cited do not adequately illustrate this claim. I note that there is no mention of a systematic review of the literature to identify publications of Phase I trials reporting on cognitive dysfunction. At the end of the manuscript (Page 13 last paragraph) you mention that a literature review was presented as a poster at the 2018 ICCTF meeting. What type of a review was this systematic? How did you identify the papers cited in the current manuscript? Suggest revising the paragraph to identify how the cited references were identified and to more accurately reflect the findings of the papers. It may be that the only appropriate citation is reference # 32.

Page 9 – re: Aims – You mention that in addition to evaluating the occurrence of cognitive dysfunction in patients receiving targeted agents in a Phase I setting there is the added aspect of feasibility to evaluate patient acceptance and appropriateness of a cognitive assessment. Other than asking patients during the qualitative portion of the study re: their experience undergoing cognitive assessment, I see no true measure of feasibility. Feasibility studies can take many forms and areas of feasibility can include measures of acceptability, demand, implementation, practicality, etc. The feasibility measure(s) are dependent on the question of interest. These measures should be planned in advance. If you wish to evaluate feasibility of conducting a battery of tests for cognitive assessment you need to stipulate up front what the feasibility measure is, what you will actually measure and a target value that would make cognitive assessment feasible, e.g. Measure - acceptability of cognitive assessment in patients enrolled in Phase I trials, Metric - proportion of patients willing to

participated in the study, Feasibility target – 85% of patients approached agreed to participate in the study. There are other feasibility endpoints to evaluate in this setting the above is just an example. If you do not plan to identify and evaluate feasibility endpoints, then you should not use the word “feasibility” but rather something like patient satisfaction/acceptance/opinion of cognitive assessment.

Page 9 – Line 58 – use of “novel therapy” – Novel therapy does not necessarily mean targeted therapy. Suggest using “targeted therapy.

Page 10 – Sample – There are no inclusion or exclusion criteria articulated. I appreciate that the sampling strategy is non-probability based; however, inclusion/exclusion criteria should still be identified. For example, since the primary focus of the study is to evaluate the cognitive effects of targeted therapies, and inclusion criteria should be: patient is scheduled to receive a targeted agent in a Phase I setting. Will you accept patients on intravenous or oral targeted therapies, perhaps both? An exclusion criteria might be: patients with uncontrolled psychiatric disorders. You mention “eligible” patients in Figure 1, how will you determine eligibility without defining the criteria in advance? Please list inclusion and exclusion criteria.

Page 10 – Your primary outcome is mean change in HVL-T-R scores between baseline and day 28 after start of therapy. This lends itself to a sample size calculation. Suggest considering a sample size calculation to identify a significant change in HVL-T-R score between baseline and day 28 of therapy.

Page 10 – Lines 13/14, 18/19 and 22-25 – Use of the word “gender” to describe males and females. The terms male and female are used to classify based on biological features i.e. sex. Gender is a social construct. As best as I can tell the paper you cite (reference #34) uses the terms male and female to describe the study group based on biological characteristics. Please replace “gender” with “sex”, as I believe you are indeed referring to the grouping of participants along biological lines.

Page 11 – Qualitative methods – There is no mention of an interview guide (although it is hinted at on Page 12 under Patient and Public Involvement). It is not sufficient to say that the interview will be guided by the results of the quantitative assessments. The guide serves to ensure that key questions related to the research are asked of the participant. Suggest developing an interview guide if you have not already done so and including the guide in an Appendix to the paper (if possible).

Pages 11 and 20 – Qualitative Methods and Table 2. It is stated that the interview will take place on day 28 after the patient has completed the quantitative assessments (85 – 97 minutes). Add an hour for the qualitative interview and the patient is at the Centre for 145-157 minutes). This all presumably occurs after the administration of the Phase I study drug which could be several hours depending on the drug and route/mode of administration. The patient could be in the Centre all day. Is this appropriate for palliative patients? You may want to consider conducting the interview on another day within 2-3 days of completing the quantitative portion.

Page 11 – Lines 30-34 – re: “tolerability of performing the structured assessments” - Since this is a feasibility endpoint, it would be appropriate to include a few questions with rating scales (e.g. Likert scale with descriptors) to determine the patient’s perception of the time required to complete the questionnaires, understandability of the questionnaires, etc. You also need to set a priori, the parameters related to these questions that would represent feasibility, e.g. 85 % of the patient population found the questionnaires could be completed in a reasonable amount of time. The study guide for the qualitative part of the study could still include questions

related to the completion of the structured questionnaires to gather more insight into the patient experience in completing them.

Pages 13-15 References

There are various errors and omissions in the citations. Here are a few that I came across during my review of the manuscript. I did not look through each citation and link. Please review the citations and ensure that the information is correct and edit as needed.

Citation #2 – Link sends reader to the pay page. A more appropriate link would be to send the reader to the main page of the publication, which is at: <https://onlinelibrary.wiley.com/doi/10.1002/9780471462422.eoct307> The reader can then determine if they have access through their library.

Citation # 3 – See comment above re; CTCAE. Advise using link to CTEP-CTCAE main page:
https://ctep.cancer.gov/protocoldevelopment/electronic_applications/ctc.htm

Citation #5 – Link is non-functioning, correct link is”
<https://journals.plos.org/plosone/article?id=10.1371/journal.pone.0108002>

Citation # 32 – not properly cited. Please correct.

Citation # 33 – the DOI link is incorrect, you have DOI:10.1200/JCO.2017.35.15-suppl.11011. The link to the abstract should be: https://ascopubs.org/doi/abs/10.1200/JCO.2017.35.15_suppl.11011

Citation # 39 – link does not work. Please provide appropriate link

Citation # 40 – link does not work. Please provide appropriate link

Page 17 – Figure 1

I found the figure somewhat confusing. As it is currently structured it would suggest that the quantitative and qualitative aspects of the trial are occurring simultaneously/in parallel rather than consecutively as the manuscript suggests. The abbreviations in the “boxes” are not defined anywhere, although one could figure them out, they should be defined. The lines joining the boxes are improperly configured, of different weights and not aligned. I would suggest deleting the figure as the description of the study methods in the manuscript is sufficient.

Page 18 - Figure 2

I found this figure somewhat confusing. I would suggest redesigning the figure to be more like a flow diagram/decision tree with the patient consenting to the Phase I trial as the first question e.g. Patient consents to Phase I trial – Yes/No. If Yes patient is approached re: Cognitive Study – patient consents to Cognitive study - Yes/No . . .etc. I am also somewhat surprised that there is are specified Day 8 and 28 post administration of trial drug as assessment points. Phase I study assessments will be driven by the type of study drug and cycle interval and may not fall on the days specified for the Cognitive Study. It may be more appropriate to just align the Cognitive Study assessments with the actual assessment times as specified for the Phase I study. This of course poses other issues related to the patients returning for the Phase I study assessments at different time points than specified for the cognitive assessment study. Alternatively you ask the patient to come in for the cognitive study assessments as planned and not align them to the Phase I study visits. This may require extra visit(s) by the patient. There are pros and cons to either approach. I feel that you will limit patient accrual if you require the design of the Phase I trial assessments to align with the Cognitive study assessments.

Page 19 – Table 1 – SQ1 – there would appear to be text missing “Sub

	Question (SQ) column - the sentence is incomplete. In the "Analysis" column – should be ". . . deterioration of cognitive function" rather than ". . . deterioration on cognitive function" Column "Sub Question (SQ)" – "RQ3" should be "SQ3". Also in the box you are using "acceptability" rather than "feasibility". These are not interchangeable terms. I note that you have also identified a measure of acceptability, "Attrition" but not defined what attrition rate would make cognitive assessment unacceptable. This relates to my comment above re: what a formal feasibility study requires. I see you have clearly thought about this. You need to expand on these endpoints in the manuscript as per my comment above. Column "Sub Question (SQ)" – "RQ4" should be "SQ4". Column re: "Methods and Data Collection" – How will the Psychologist assess "accuracy" of the nurse performed assessment. Will the Psychologist have the patient complete the questionnaires a second time? This would likely impact on the patient's perception of conducting cognitive assessments. How the Psychologist is to assess "accuracy" needs to be further described in the manuscript.
--	--

VERSION 1 – AUTHOR RESPONSE

Thank you for your constructive feedback on the protocol manuscript. I have hopefully addressed these comments and edited appropriately. I have uploaded a clean version of the new document as well as a version with changes in bold for ease of review. Please see below responses to individual comments. I look forward to hearing your response to the resubmission.

The study lacks a control group and will use a convenience sample; both of these decisions are justified in the context of the Phase I trial. However, I think the selection of primary and secondary endpoints does require additional explanation. Please explain why the HVLIT was chosen as the primary endpoint, over the other objective neuropsychological tests. Also, why is change from baseline to day 8 a secondary and not primary endpoint?

Since initially submitting this protocol more work has been conducted on the analysis plan. Some thought was given to the framing of the study. As it is exploratory it was felt by the supervisory and statistics support that the primary and secondary endpoints should not be stated and that exploratory analysis would aim to describe the scores at each timepoint and describe changes over times, testing the methods and identifying an effect size for future power calculations. Rather than endpoints for hypothesis testing which was decided was not really appropriate for the sample size. The new analysis plan has been inserted on page 13 under the heading statistical analysis plan.

Table 1: SQ1 question is unfinished

Table 1 has been corrected now Thankyou

Random odd capitalization choices throughout the manuscript, e.g. on p. 5 "In Cancer drug...", table 1:

"...in

Psychometric testing..."

Thankyou I have proof read the manuscript and hopefully all grammar is corrected.

p. 11, Qualitative methods: I hope that by "illicit information" you mean "elicit information"

Yes this I corrected now.

p. 12 “The primary endpoint will be analysed in the first instance using a

paired t-test if the data are normally distributed, if not, a non-parametric method such as the Wilcoxon matched pairs signed rank test will be used.”

Consider using the Bayesian t-test, which allows you to compare evidence for and against the null hypothesis.

As above the new analysis plan is now inserted on page 13.

Reviewer: 2

Dr. Carlo DeAngelis, Sunnybrook Health Sciences Centre, University of Toronto Leslie Dan Faculty of Pharmacy

Comments to the Author:

I found your research idea of interest and believe that there is a need to more effectively monitor patients enrolled in clinical trials for side effects, not just for cognitive dysfunction but for other side effects as well. There needs to be a more proactive rather than the current reactive approach which relies on the patient to report side effects sometimes weeks after receiving therapy. Overall the mixed methods approach is appropriate. I do have some concerns listed below. It may be that you have already thought of the issues I outline but have not incorporated the detail in the manuscript. However, I can only comment on what I have available to me to read. I structured my comments in the order in which they arose during the reading of the manuscript. These include minor edits and more substantive questions/concerns.

Page 4 - Lines 34/35 – end of sentence – I believe “for” should be deleted, for the sentence to read properly.

Page 4 - Lines 37/38 – beginning of the sentence – I believe “is” should be deleted, for the sentence to read properly

Page 4 - Line 59/60 – mid-sentence – “form” should be “from”

Page 5 – Line 15 – mid-sentence – “to” should be

“do” **These edits have been corrected now**

Thankyou

Page 5 – Lines 38-41 – Sentence re: CTCAE – the Common Terminology Criteria for Adverse Events (CTCAE) is a National Institutes of Health – National Cancer Institute, Division of Cancer

Treatment and Diagnosis, Cancer Therapy Evaluation Program initiative not an EORTC initiative. Please correct the sentence with respect to the source of the CTCAE. Also use the correct designation: Common Terminology Criteria for Adverse Events (CTCAE). Finally re: citation #3 – The link does not work. Rather than direct the reader to a PDF version of the NCI-CTCAE suggest using the main page link:

https://ctep.cancer.gov/protocoldevelopment/electronic_applications/ctc.htm

I have updated this sentence and inserted the correct link in the references

Page 5 – Lines 47-49 – Sentence – “This exploratory study investigates the impact on the cognitive function of these patients with advanced cancer, who are participating in Phase I trials.”

Grammatically is missing the impact of “what”. Suggest: “This exploratory study investigates the impact of novel targeted therapies on the cognitive function of patients with advanced cancer, who are participating in Phase I trials.”

I have expanded this sentence to include novel targeted therapies

Page 7 – Lines 47-54 – This is a reasonable justification as to why a group of control patients will not be recruited into the study. I would also suggest that given that this is an exploratory study, not having a control group is also consistent with the study’s objectives. The ICCTF also suggests that a control group is not always feasible and that the control group will depend on the research question under study.

Thankyou

Page 8 – Line 5 – re: citing of statement made in the sentence – “19,20,21,22,23,24,25,26” - Multiple citations that are consecutively numbered can be cited as a range i.e. 19-26

This has been rectified as advised thankyou

Page 8 Lines 32-37 – re: Definition of Dose Limiting Toxicity – Please revise the sentence to include that the occurrence of a DLT prevents further dose escalation of the study drug. See NIH-NCI definition at: <https://www.cancer.gov/publications/dictionaries/cancer-terms/def/dose-limiting>

I have updated the sentence with the more comprehensive definition of a DLT as advised thankyou.

Page 8 – Lines 37-57 - I could not verify the information cited (references 30-33).

- Reference # 30 (Bendell JC, et al.) – BKM120 was given as a single agent not in combination with an mTOR inhibitor as suggested in the manuscript text. In addition in the Bendell paper, I could not find any mention in the text of “memory loss, confusion, or attention deficit and agitation”. They reported mood effects and anxiety. Terms included in mood effects, included “mood altered, anxiety, depression, emotional disorder, crying episodes, hallucinations and irritability”. The paper reports all Grade “Mood altered” as 20%, (Table 2). Anxiety was reported separately (17% Table 2). In Table 2 the study

population size was 35, not 28 as suggested in this manuscript. I did not have access to the supplemental material for the Bendell paper where there may have been corroborating information supporting your statement re” the Bendell paper. I am also concerned with the use of “Mood Effects” being conflated as cognitive dysfunction. This is not consistent with your definition/description of cognitive dysfunction on Page 6 of the manuscript (mid-page).

- Reference #31 – (Fong PC, et al.) – I could not find any mention of mood effects or any other CNS effects in the toxicity table (Table 3). In text they describe two cases of DLT due to altered mood and fatigue (1 patient) and another patient with extreme somnolence. Here again I am concerned about conflating the mentioned side effects with cognitive dysfunction.

- Reference # 32 – (McKay RR, et al.) In this paper they appear to make a distinction between “cognitive disturbance” and “mood effects”. They used CTCAE v 4.0 to define this endpoint. There was one patient who had a DLT related to cognitive dysfunction. This paper is your best example of what you are claiming, i.e. cognitive dysfunction is not systematically assessed and because assessment relies on patient reporting the adverse effect, it may be missed. I also note that in this paper baparisib (= BKM120 from citation # 30) is used in combination. Did you mean to refer to this paper earlier in the paragraph when citing reference #30 as a study of BKM120 given in combination?

- Reference #33 – (Heinrich MC, et al. Abstract) – There is no mention of mood effects or cognitive dysfunction.

Overall I found this paragraph to not accurately reflect the information in the publications cited. I do not disagree with your assertion that cognitive dysfunction is not commonly reported in Phase I trial. I do however, have some concern that the publications cited do not adequately illustrate this claim. I note that there is no mention of a systematic review of the literature to identify publications of Phase I trials reporting on cognitive dysfunction. At the end of the manuscript (Page 13 last paragraph) you mention that a literature review was presented as a poster at the 2018 ICCTF meeting. What type of a review was this systematic? How did you identify the papers cited in the current manuscript? Suggest revising the paragraph to identify how the cited references were identified and to more accurately reflect the findings of the papers. It may be that the only appropriate citation is reference # 32.

Apologies to the lack of accuracy for some of this referencing, some if the evidence is cited incorrectly. The scoping literature review has been re-run , and the manuscript updated to better reflect the evidence. There are some Phase I papers highlighting cognitive impairment as toxicity and the point I was attempting to make is that the symptom is never well characterised and recorded as a category of cognitive symptoms rather than defined as a symptom. I hope that the papers cited better support that need for research aiming to improve the assessments in this area. The updated scoping review begins on pg 6 (clean version) of the edited manuscript.

Page 9 – re: Aims – You mention that in addition to evaluating the occurrence of cognitive dysfunction in patients receiving targeted agents in a Phase I setting there is the added aspect of feasibility to evaluate patient acceptance and appropriateness of a cognitive assessment. Other than asking patients during the qualitative portion of the study re: their experience undergoing cognitive assessment, I see no true measure of feasibility. Feasibility studies can take many forms and areas of feasibility can include measures of acceptability, demand, implementation, practicality, etc. The feasibility measure(s) are dependent on the question of interest. These measures should be planned in advance. If you wish to evaluate feasibility of conducting a battery of tests for cognitive assessment you need to stipulate up front what the feasibility measure is, what you will actually measure and a

target value that would make cognitive assessment feasible, e.g. Measure - acceptability of cognitive assessment in patients enrolled in Phase I trials, Metric - proportion of patients willing to participated in the study, Feasibility target – 85% of patients approached agreed to participate in the study. There are other feasibility endpoints to evaluate in this setting the above is just an example. If you do not plan to identify and evaluate feasibility endpoints, then you should not use the word “feasibility” but rather something like patient satisfaction/acceptance/opinion of cognitive assessment.

Feasibility points have been identified in terms of delivery and acceptability and will be measures as uptake of trial, attrition rates, reasons for non-uptake and frequency of positive responses from qualitative data. Coding and narrative of qual data. Accuracy of nurse measurements will be assessed in terms of delivery through supervision/observation of early assessments, and checking the duplicate checking of recording and interpretation of recorded tests,. The patients will not be expected to take the same test twice to conduct inter rater reliability. This has now been updated in the manuscript in table 1 and in the analysis section. Page 9 – Line 58 – use of “novel therapy” – Novel therapy does not necessarily mean targeted therapy. Suggest using “targeted therapy.

This is amended

Page 10 – Sample – There are no inclusion or exclusion criteria articulated. I appreciate that the sampling strategy is non-probability based; however, inclusion/exclusion criteria should still be identified. For example, since the primary focus of the study is to evaluate the cognitive effects of targeted therapies, and inclusion criteria should be: patient is scheduled to receive a targeted agent in a Phase I setting. Will you accept patients on intravenous or oral targeted therapies, perhaps both? An exclusion criteria might be: patients with uncontrolled psychiatric disorders. You mention “eligible” patients in Figure 1, how will you determine eligibility without defining the criteria in advance? Please list inclusion and exclusion criteria.

Inclusion/exclusion criteria are inserted on pg 10 (clean)

Page 10 – Your primary outcome is mean change in HVLT-R scores between baseline and day 28 after start of therapy. This lends itself to a sample size calculation. Suggest considering a sample size calculation to identify a significant change in HVLT-R score between baseline and day 28 of therapy.

Since initially submitting this protocol more work has been conducted on the analysis plan. Some thought was given to the framing of the study. As it is exploratory it was felt by the supervisory and statistics support that the primary and secondary endpoints should not be stated and that exploratory analysis would aim to describe the scores at each timepoint and describe changes over times, testing the methods and identifying an effect size for future power calculations. Rather than endpoints for hypothesis testing which was decided was not really appropriate for the sample size. The new analysis plan has been inserted on page 13 under the heading statistical analysis plan.

Page 10 – Lines 13/14, 18/19 and 22-25 – Use of the word “gender” to describe males and females. The terms male and female are used to classify based on biological features i.e. sex. Gender is a social construct. As best as I can tell the paper you cite (reference #34) uses the terms male and

female to describe the study group based on biological characteristics. Please replace “gender” with “sex”, as I believe you are indeed referring to the grouping of participants along biological lines.

This has been corrected as advised

Page 11 – Qualitative methods – There is no mention of an interview guide (although it is hinted at on Page 12 under Patient and Public Involvement). It is not sufficient to say that the interview will be guided by the results of the quantitative assessments. The guide serves to ensure that key questions related to the research are asked of the participant. Suggest developing an interview guide if you have not already done so and including the guide in an Appendix to the paper (if possible).

Apologies for not included the interview guide this is now included as appendix A. It was developed with PPI input and has been approved through the ethics submission process.

Pages 11 and 20 – Qualitative Methods and Table 2. It is stated that the interview will take place on day 28 after the patient has completed the quantitative assessments (85 – 97 minutes). Add an hour for the qualitative interview and the patient is at the Centre for 145-157 minutes). This all presumably occurs after the administration of the Phase I study drug which could be several hours depending on the drug and route/mode of administration. The patient could be in the Centre all day. Is this appropriate for palliative patients? You may want to consider conducting the interview on another day within 2-3 days of completing the quantitative portion.

Thankyou for raising this thoughtful concern A lot of thought went into the best way to manage this and it was decided in conjunction with the patient groups. The drug development unit that the study is conducted in has portfolio off over 40 Phase I trials running at any one time. All trials have weekly visits so day 1,8,15,22 and so on. The patients (some of whom travel quite far to go to the centre) spend most of the day there and have long periods in between assessments/administration of dose. Therefore it was decided that to fit the assessments in between (during waiting times) would be more convenient for the patients. During recruitment we have found that this works well and the patients prefer it to an extra visit.

Page 11 – Lines 30-34 – re: “tolerability of performing the structured assessments” - Since this is a feasibility endpoint, it would be appropriate to include a few questions with rating scales (e.g. Likert scale with descriptors) to determine the patient’s perception of the time required to complete the questionnaires, understandability of the questionnaires, etc. You also need to set a priori, the parameters related to these questions that would represent feasibility, e.g. 85 % of the patient population found the questionnaires could be completed in a reasonable amount of time. The study guide for the qualitative part of the study could still include questions related to the completion of the structured questionnaires to gather more insight into the patient experience in completing them.

See point above re feasibility

Pages 13-15 References

There are various errors and omissions in the citations. Here are a few that I came across during my review of the manuscript. I did not look through each citation and link. Please review the citations and ensure that the information is correct and edit as needed.

Citation #2 – Link sends reader to the pay page. A more appropriate link would be to send the reader to the main page of the publication, which is at:

<https://onlinelibrary.wiley.com/doi/10.1002/9780471462422.eoct307> The reader can then determine if they have access through their library.

Citation # 3 – See comment above re; CTCAE. Advise using link to CTEP-CTCAE main page:

https://ctep.cancer.gov/protocoldevelopment/electronic_applications/ctc.htm

Citation #5 – Link is non-functioning, correct link is”

<https://journals.plos.org/plosone/article?id=10.1371/journal.pone.0108002>

Citation # 32 – not properly cited. Please correct.

Citation # 33 – the DOI link is incorrect, you have DOI:10.1200/JCO.2017.35.15-suppl.11011. The link to the abstract should be: https://ascopubs.org/doi/abs/10.1200/JCO.2017.35.15_suppl.11011

Citation # 39 – link does not work. Please provide appropriate link

Citation # 40 – link does not work. Please provide appropriate link

I have checked and updated the citations thank you for highlighting the inaccuracies.

Page 17 – Figure 1

I found the figure somewhat confusing. As it is currently structured it would suggest that the quantitative and qualitative aspects of the trial are occurring simultaneously/in parallel rather than consecutively as the manuscript suggests. The abbreviations in the “boxes” are not defined anywhere, although one could figure them out, they should be defined. The lines joining the boxes are improperly configured, of different weights and not aligned. I would suggest deleting the figure as the description of the study methods in the manuscript is sufficient.

I have removed this figure as suggested

Page 18 - Figure 2

I found this figure somewhat confusing. I would suggest redesigning the figure to be more like a flow diagram/decision tree with the patient consenting to the Phase I trial as the first question e.g. Patient consents to Phase I trial – Yes/No. If Yes patient is approached re: Cognitive Study – patient consents to Cognitive study - Yes/No . . .etc. I am also somewhat surprised that there are specified Day 8 and 28 post administration of trial drug as assessment points. Phase I study assessments will be driven by the type of study drug and cycle interval and may not fall on the days specified for the Cognitive Study. It may be more appropriate to just align the Cognitive Study assessments with the actual assessment times as specified for the Phase I study. This of course poses other issues related to the patients returning for the Phase I study assessments at different time points than specified for the cognitive assessment study. Alternatively you ask the patient to come in for the cognitive study assessments as

planned and not align them to the Phase I study visits. This may require extra visit(s) by the patient. There are pros and cons to either approach. I feel that you will limit patient accrual if you require the design of the Phase I trial assessments to align with the Cognitive study assessments.

I have kept the plan of the assessments as is as it has been approved by ethics for this trial to run concurrently with the Phase I on the set timepoints.

Page 19 – Table 1 – SQ1 – there would appear to be text missing “Sub Question (SQ)” column - the sentence is incomplete. In the “Analysis” column – should be “. . . deterioration of cognitive function” rather than “. . . deterioration on cognitive function”

Column “Sub Question (SQ)” – “RQ3” should be “SQ3”. Also in the box you are using “acceptability” rather than “feasibility”. These are not interchangeable terms. I note that you have also identified a measure of acceptability, “Attrition” but not defined what attrition rate would make cognitive assessment unacceptable. This relates to my comment above re: what a formal feasibility study requires. I see you have clearly thought about this. You need to expand on these endpoints in the manuscript as per my comment above.

Column “Sub Question (SQ)” – “RQ4” should be “SQ4”. Column re: “Methods and Data Collection” – How will the Psychologist assess “accuracy” of the nurse performed assessment. Will the Psychologist have the patient complete the questionnaires a second time? This would likely impact on the patient’s perception of conducting cognitive assessments. How the Psychologist is to assess “accuracy” needs to be further described in the manuscript.

I have rewritten table 1 to better describe the research questions with aims aligned with methods of assessment and analysis. Hopefully this is now more meaningful. Thankyou.

VERSION 2 – REVIEW

REVIEWER	Horowitz, Todd National Institutes of Health, National Cancer Institute
REVIEW RETURNED	15-Feb-2022

GENERAL COMMENTS	My only substantial comment on the previous draft was that I did not understand the justification for the primary and secondary endpoints. The authors have re-framed the study as an exploratory study, so that question becomes moot. However, the new analysis section still seems incomplete. The authors write: “The proportions of patients showing CI (as defined in the methods of data collection section) will be reported as percentages.” However, the definition of CI has been left out of the “methods of data collection section”. Indeed, there is no such section. There is a “methods” section, and a “methods of data collection” table (which does not define CI), but no “methods of data collection section”. In fact, I am just guessing that “CI” refers to “cognitive impairment”, since it is defined nowhere in the text. This is a critical piece of information. As Shilling et al. have shown [Shilling, V., Jenkins, V., & Trapala, I. S. (2005). The (mis)classification of chemo-fog – methodological inconsistencies in the investigation of cognitive impairment after chemotherapy. Breast Cancer Research and Treatment , 95(2), 125–129. https://doi.org/10.1007/s10549-005-9055-1], varying definitions for “impairment” is a major source of apparent disagreement across the cancer-related cognitive impairments literature.
---

	My guess is that a section or paragraph on this was written and omitted by accident. This feeds into my other concern, which normally be fairly minor: lack of attention to detail. Despite the response letter's assurance that "I have proof read the manuscript and hopefully all grammar is corrected," I found errors on nearly every page. Most were fairly trivial, of course (odd capitalization, unfinished sentences, reduplication), but it does reduce my confidence that all of the critical information has been included in the manuscript. If the definition of cognitive impairment is missing, what else was accidentally left out? I normally do not provide detailed proofreading comments, but I think when we're dealing with a protocol, attention to detail is critical. Here is a sampling of items I found in a cursory read-through. p. 2: "This could lead to toxicity profiles treatments not being fully described..." What is meant here? Toxicity profiles not being fully described? Toxicity profiles of treatments not being fully described? "30 Patients". Should be "thirty patients". p. 5: "(13,14,)." "... matched healthy controls. (18 -20)." p. 8: "...in particular the Patient reported outcomes..." "A similar study using the MD Anderson using the MDAnderson symptom inventory..." "... which is a also patient reported questionnaire..." "... in the PROM's studies..." "Secondly patients who are participating in Phase I trials and may have unmet need in terms of identification and support of symptoms of cognitive impairment. " note: what is the "and" doing there? Table 1 continues to refer to PROMs as "PROM's" or "PROM". Table 1, RQ 5: "Attrition rates recorded and Qualitative enquiry as part of the interview to ask how acceptable the cognitive assessment battery has been" Unclear if this ("attrition rates recorded and...") is a sentence fragment or if there's supposed to be a bridge to "Qualitative enquiry" p. 12: Table 2: methods of data collection Usually you don't include the title of the table when you're referencing it in text. p. 14: "Changes over time of perceived impairment will be evaluated using the established MCID of 9 points difference form baseline will be presented"
--	--

I think you mean “from” not “form”, and it’s not clear what “will be presented” is doing there.

VERSION 2 – AUTHOR RESPONSE

Thank you for your constructive follow up comments. I have amended as below and completed a further proofread. The responses to your individual comments are in bold. I have changes to title to meet your requirements as outline in the response email.

My only substantial comment on the previous draft was that I did not understand the justification for the primary and secondary endpoints. The authors have re-framed the study as an exploratory study, so that question becomes moot. However, the new analysis section still seems incomplete. The authors write: “The proportions of patients showing CI (as defined in the methods of data collection section) will be reported as percentages.” However, the definition of CI has been left out of the “methods of data collection section”. Indeed, there is no such section. There is a “methods” section, and a “methods of data collection” table (which does not define CI), but no “methods of data collection section”. In fact, I am just guessing that “CI” refers to “cognitive impairment”, since it is defined nowhere in the text. This is a critical piece of information. As Shilling et al. have shown [Shilling, V., Jenkins, V., & Trapala, I. S. (2005). The (mis)classification of chemo-fog – methodological inconsistencies in the investigation of cognitive impairment after chemotherapy. *Breast Cancer Research and Treatment*, 95(2), 125–129. <https://doi.org/10.1007/s10549-005-9055-1>], varying definitions for “impairment” is a major source of apparent disagreement across the cancer-related cognitive impairments literature.

Thankyou I have added in clinically significant classification of Cognitive Impairment (CI) in the analysis section. See sentence below. I have also initialised the first cognitive impairment in the text and referred to CI from there on.

For the HVLt-R memory test CI will be defined as a T score of less than 42 (43). For the CTMT A and B CI is defined as a T score less than 40 and for the COWA CI is defined as a Z score less than -1 (44).

My guess is that a section or paragraph on this was written and omitted by accident. This feeds into my other concern, which normally be fairly minor: lack of attention to detail. Despite the response letter’s assurance that “I have proof read the manuscript and hopefully all grammar is corrected,” I found errors on nearly every page. Most were fairly trivial, of course (odd capitalization, unfinished sentences, reduplication), but it does reduce my confidence that all of the critical information has been included in the manuscript. If the definition of cognitive impairment is missing, what else was accidentally left out?

I normally do not provide detailed proofreading comments, but I think when we’re dealing with a protocol, attention to detail is critical. Here is a sampling of items I found in a cursory read-through.

p. 2: “This could lead to toxicity profiles treatments not being fully described...”
What is meant here? Toxicity profiles not being fully described? Toxicity profiles of treatments not being fully described?

This has been amended as below now

This could lead to toxicity profiles of new treatments not being fully described and patients with unmet need in terms of acknowledgement and support of symptoms.

“30 Patients”. Should be “thirty patients”.

This has been changed to thirty

p. 5: "(13,14,)." **(13,14).**

"... matched healthy controls. (18 -20)."
compared to matched healthy **controls (18 -20).**

p. 8: "...in particular the Patient reported outcomes..."

Capitalisation corrected and initialised as PROMs throughout the text now

"A similar study using the MD Anderson using the
MDAnderson symptom inventory..."

"... which is a also patient reported questionnaire..."

A similar study using the MD Anderson symptom inventory (MDASI) which is also a patient reported questionnaire

"... in the PROM's studies..."

Corrected to PROMs

"Secondly patients who are participating in Phase I trials and
may have unmet need in terms of identification and support of symptoms of cognitive impairment."
"

note: what is the "and" doing there?

Secondly patients who are participating in phase I trials may have unmet need in terms of identification and support of symptoms of CI.

Table 1 continues to refer to PROMs as "PROM's" or "PROM".

Corrected throughout

Table 1, RQ 5: "Attrition rates recorded and
Qualitative enquiry as part of the
interview to ask how acceptable
the cognitive assessment battery
has been"

Unclear if this ("attrition rates recorded and...") is a sentence fragment or if there's supposed to be a bridge to "Qualitative enquiry"

These are two separate ways to measure feasibility and have been separated in the table now.

p. 12: Table 2: methods of data collection
Usually you don't include the title of the table when you're referencing it in text.

Thank you have removed this now

p. 14: "Changes over time of perceived impairment will be evaluated using the established MCID of 9 points difference form baseline will be presented"

I think you mean "from" not "form", and it's not clear what "will be presented" is doing there.

Thankyou for pointing out these errors I have amended them now.

VERSION 3 – REVIEW

REVIEWER	Horowitz, Todd National Institutes of Health, National Cancer Institute
REVIEW RETURNED	23-Jul-2022

GENERAL COMMENTS	Manuscript is now acceptable. I would suggest another round of proofreading, but I found no content issues. p 4.: "anecdotal reports from some patients suggest that they experience some deterioration whilst on the trial which is often temporal to dosing of drug. " Meaning of "temporal" is unclear.
--